# Prognostic factors for persistent obstructive symptoms in patients with Hirschsprung disease following pull-through

**Naisya Balela, Aditya Rifqi Fauzi, Ninditya Nugroho, Andi Dwihantoro, Gunadi** *

Faculty of Medicine, Department of Surgery, Pediatric Surgery Division, Public Health and Nursing, Universitas Gadjah Mada/Dr. Sardjito Hospital, Yogyakarta, Indonesia

* drgunadi@ugm.ac.id

## Abstract

### Background

Although most patients with Hirschsprung disease (HSCR) improve after pull-through, some patients still have persistent obstructive symptoms. Most previous studies reported persistent obstructive symptoms after pull-through in HSCR patients from developed countries. Our study determined the prognostic factors of persistent obstructive symptoms in patients with HSCR following pull-through from a particular developing country.

### Methods

A cross-sectional study was conducted using medical records of patients with HSCR at our institution from January 2017 to January 2022.

### Results

We ascertained 114 patients with HSCR: 79 males and 35 females. Most of them (90.4%) showed a short aganglionosis and underwent transanal endorectal pull-through (55.3%). Twenty-two percent of patients showed persistent obstructive symptoms following pull-through. Operative technique and age at definitive surgery were significantly associated with the persistent obstructive symptoms after pull-through ($p = 0.011$ and $0.019$, respectively), while sex, aganglionic segment length, presence of global developmental delay, and Down syndrome were not ($p = 0.873$, $0.525$, $0.647$, and $0.301$, respectively). Multivariate analysis revealed that age at pull-through was a significant independent factor for persistent obstructive symptoms after pull-through, with an odds ratio of 3.41 (95% CI = 1.18–9.91; $p = 0.02$).

### Conclusions

Our study shows a moderate frequency of persistent obstructive symptoms after pull-through in our institution. In addition, patients who underwent pull-throughs at a younger age might have persistent obstructive symptoms following a definitive surgery. Our study provides new data on persistent obstructive symptoms after pull-through from a particular

**Data Availability Statement:** All relevant data are within the paper and its Supporting Information files.

**Funding:** The author(s) received no specific funding for this work.

**Competing interests:** The authors have declared that no competing interests exist.

population that might be beneficial for pediatric surgeons' consideration before performing definitive surgery on patients with HSCR.

## Introduction

Hirschsprung disease (HSCR) is a complex hereditary disorder characterized by the lack of an enteric ganglion in the submucosa and a myenteric plexus that varies along the distal intestine. HSCR affects 1 in every 5,000 live births, with the Asian population having the most significant incidence, 2.8 out of 10,000 live births. HSCR shows a male: female ratio of 4:1 [1]. Its incidence in Indonesia is approximately 1:3,250 [2].

Although most HSCR patients improved after surgery, some patients suffered from the possibility of postoperative complications [3], including persistent obstructive symptoms following pull-through [3,4]. Postoperative persistent obstructive symptoms were defined as abdominal distension, bloating, borborygmi, vomiting, or severe constipation following pull-through [3]. The incidence of postoperative persistent obstructive symptoms is approximately 8–30% [1]. Moreover, persistent obstructive symptoms after pull-through can lead to chronic enterocolitis, resulting in failure to thrive in children [5]. Most previous studies reported persistent obstructive symptoms after pull-through in HSCR patients from developed countries [3–5]. Here, we aimed to determine the prognostic factors of persistent obstructive symptoms in patients with HSCR from a particular developing country.

## Material and methods

### Patients

A cross-sectional study was conducted using medical records of patients with HSCR at Dr. Sardjito Hospital, Yogyakarta, Indonesia, from January 2017 to January 2022. Diagnosis of HSCR in our hospital was established according to histopathological findings. HSCR patients were classified as short-segment (aganglionosis affects the rectosigmoid colon), long-segment (aganglionosis extending proximal to the splenic flexure), and total colon aganglionosis (TCA, aganglionosis of the entire colon). The aganglionic segment length was determined using intraoperative histopathological evaluations (frozen sections) [6]. Intraoperative frozen sections were compared with permanent sections to ensure the segment length was appropriately diagnosed. All frozen sections samples, except two cases, correlated with the permanent sections.

The inclusion criterion was all HSCR patients <18 years old and underwent pull-through in our institution. The exclusion criteria were deceased patients and incomplete medical records. We included 114 HSCR patients with persistent obstructive symptoms after the transanal endorectal (TEPT), Swenson-like, and Duhamel pull-through. The data were retrospectively collected from March to April 2022, while the data were accessed for research purposes from April to May 2022. Authors had access to identifying information during or after data collection. The patients were taken care of at the ward at least five days after the pull-through. Subsequently, the patients were routinely follow-up at the outpatient clinic at least every two weeks until at least six months after the pull-through.

The Ethical Committee of the Faculty of Medicine, Universitas Gadjah Mada/Dr. Sardjito Hospital, Indonesia, approved the study (KE/FK/0135/EC/2022). Written informed consent was obtained from all parents for participating in this study. The research has been performed following the Declaration of Helsinki.

### Prognostic factors

Postoperative persistent obstructive symptoms were defined as abdominal distension, bloating, borborygmi, vomiting, or severe constipation following pull-through [3]. After admission, the patients with persistent obstructive symptoms were managed with antibiotics, irrigations, and nothing per oral. They were evaluated using the following diagnostic approach: history, abdominal X-ray, contrast enema, and examination under anesthesia [3,4]. Treatment of patients was determined according to the etiology of the obstructive symptoms, anatomic, pathological, or functional [3,4].

Next, we evaluated prognostic factors for postoperative persistent obstructive symptoms: sex, aganglionic segment length, operative technique, age at definitive surgery, presence of global developmental delay, and Down syndrome. The operative technique consisted of TEPT, transanal Swenson-like, and Duhamel pull-through. The pull-through methods were chosen according to the attending pediatric surgeon's discretion. To minimize bias, the authors did not know who performed the pull-through during data analysis. According to a previous study, the age at definitive surgery was classified into <1 year and ≥1 year [7].

### Definitive surgery

Prior to definitive surgery, the children were managed by rectal irrigation. If the rectal irrigation was ineffective in keeping the colon decompressed, the children underwent colostomy.

### TEPT

We put the everting sutures throughout the anus to show the mucosa. Subsequently, we incised the mucosa circumferentially about 0.5 cm above the dentate line by a needle tip cautery. We conducted a submucosal dissection proximally for about 1 cm and converted it to the full thickness of the rectal wall until the transition zone was reached. Once we confirmed the ganglion cells in the colon by frozen section, we removed at least an additional 5 cm of the colon to ensure that we resected the transition zone as well. Finally, we conducted coloanal anastomosis [8].

**Duhamel pull-through.** We performed a two-staged Duhamel pull-through for our patients in our hospital, i.e., all patients received stoma first. We made a hockey-stick incision incorporation colostomy. We mobilized the colon proximal to the prior colostomy. We then created a retrorectal space and preserved around 4 cm of the original rectum. We closed the rectal stump from the abdomen using interrupted sutures. After the posterior rectal wall was exposed, we made a full-thickness incision 0.5 cm above the dentate line. Subsequently, we pulled the mobilized colon through this incision. We performed the coloanal anastomosis using interrupted sutures. Eventually, we used an automatic stapling device to obliterate the septum between the original rectum and the pulled colon [9].

All TCA patients have been managed with a diverting ileostomy first. They underwent definitive surgery using the Duhamel approach at the age of at least one-year-old (Table 1).

### Transanal Swenson-like pull-through

We put the lone star retractor hooks just above the dentate line to hide and protect the dentate line during the dissection. We placed interrupted sutures circumferentially 1.0 cm above the dentate line. Subsequently, we conducted a full-thickness dissection using a needle-tip cautery until reaching the peritoneal cavity. We then pulled through the colon to the appropriate level. After we confirmed the ganglion cells in the colon by intraoperative histopathological findings, we resected a minimum of an additional 5 cm of the colon to ensure we resected the transition zone too. Ultimately, we performed coloanal anastomosis [10].

**Table 1. Baseline characteristics of patients with HSCR involved in this study.**

| Characteristics | N (%) |
|---|---|
| Sex | |
| ■ Male | 79 (69.3) |
| ■ Female | 35 (30.7) |
| Aganglionosis type and definitive surgery | |
| ■ Short | 103 (90.4) |
| √ TEPT | 60 |
| √ Duhamel | 30 |
| √ Transanal Swenson-like | 13 |
| ■ Long | 9 (7.9) |
| √ TEPT | 3 |
| √ Duhamel | 6 |
| √ Transanal Swenson-like | 0 |
| ■ Total colon aganglionosis | 2 (1.7) |
| √ TEPT | 0 |
| √ Duhamel | 2 |
| √ Transanal Swenson-like | 0 |
| Definitive surgery and age at pull-through (year) | |
| ■ TEPT | 63 (55.3) |
| √ <1 | 54 |
| √ ≥1 | 9 |
| ■ Duhamel | 38 (33.3) |
| √ <1 | 4 |
| √ ≥1 | 34 |
| ■ Transanal Swenson-like | 13 (11.4) |
| √ <1 | 10 |
| √ ≥1 | 3 |
| Age at pull-through (year) | |
| ■ <1 | 68 (59.6) |
| ■ ≥1 | 46 (40.4) |
| Global developmental delay | |
| ■ Yes | 7 (6.1) |
| ■ No | 107 (93.9) |
| Down syndrome | |
| ■ Yes | 5 (4.4) |
| ■ No | 109 (9.6) |
| Persistent obstructive symptoms | |
| ■ Yes | 25 (22) |
| ■ No | 89 (78) |

TEPT, transanal endorectal pull-through.

## Post pull-through management

The nasogastric tube was removed soon after surgery finished, while the feedings were given once the bowel sound was normal. The Foley catheter was removed on postoperative day (POD) 2 after Duhamel and TEPT, and POD4 after transanal Swenson-like pull-through. Antibiotics were given for five days after surgery. Most patients were discharged on POD5.

## Statistical analysis

The Chi-square or Fisher Exact test was utilized to assess the association between prognostic variables and persistent obstructive symptoms after pull-through. The multivariate analysis was conducted using the backward stepwise logistic regression to identify significant prognostic factors for persistent obstructive symptoms after pull-through. A $p$-value of <0.05 was considered to be significant. IBM SPSS Statistics version 20 (SPSS Chicago, USA) was used for all statistical analyses.

**Table 2. Association between prognostic factors and persistent obstructive symptoms following pull-through in HSCR patients.**

| Factor Prognostics | Persistent obstructive symptoms | | p-value | OR (95% CI) |
|---|---|---|---|---|
| | Yes (N, %) | No (N, %) | | |
| Sex | | | | |
| ▪ Male | 17 (14.9) | 62 (54.4) | 0.862 | 0.93 (0.36–2.40) |
| ▪ Female | 8 (7) | 27 (23.7) | | |
| Aganglionosis type | | | | |
| ▪ Short | 24 (21.1) | 79 (69.3) | 0.469 | 0.41 (0.05–3.46) |
| ▪ Long | 1 (0.9) | 8 (7) | 1 | N/A |
| ▪ Total colon aganglionosis | 0 | 2 (1.8) | | |
| Definitive surgery | | | | |
| ▪ TEPT | 19 (16.7) | 44 (38.6) | 0.011* | 0.20 (0.05–0.73) |
| ▪ Duhamel | 3 (2.6) | 35 (30.7) | 0.745 | 0.69 (0.17–2.81) |
| ▪ Transanal Swenson-like | 3 (2.6) | 10 (8.8) | | |
| Age at pull-through (year) | | | | |
| ▪ <1 | 20 (17.5) | 48 (42.1) | 0.019 | 3.42 (1.18–9.91) |
| ▪ ≥1 | 5 (4.4) | 41 (36) | | |
| Global developmental delay | | | | |
| ▪ Yes | 2 (1.8) | 5 (4.4) | 0.647 | 1.46 (0.27–8.02) |
| ▪ No | 23 (20.2) | 84 (73.7) | | |
| Down syndrome | | | | |
| ▪ Yes | 2 (1.8) | 3 (2.6) | 0.584 | 2.49 (0.39–15.81) |
| ▪ No | 23 (20.2) | 86 (75.4) | | |

*, $p<0.05$; CI, confidence interval; OR, odds ratio; HSCR, Hirschsprung disease; N/A, not applicable; TEPT, transanal endorectal pull-through.

## Results

### Baseline characteristics

We involved 114 HSCR patients. Most of them had short aganglionosis (90.4%), underwent TEPT (55.3%), and had age at surgery < 1-year-old (59.6%) (Table 1). Twenty-five (22%) patients showed persistent obstructive symptoms following pull-through. Time to the incidence of obstructive symptoms varied among patients from 10 days to 16 months following pull-through (S1 Table).

### Association between prognostic factors and persistent obstruction of children with HSCR

Operative technique and age at definitive surgery were significantly associated with persistent obstructive symptoms after pull-through ($p$ = 0.011 and 0.019, respectively). At the same time, sex, aganglionic segment length, presence of global developmental delay, and Down syndrome were not (Table 2).

### Multivariate analysis of prognostic factors for persistent obstruction of children with HSCR

Next, we performed a multivariate analysis. Age at pull-through was a significant independent factor for persistent obstructive symptoms after pull-through, with an odds ratio ((OR) of 3.41 (95% CI = 1.18–9.91; $p$ = 0.02) (Table 3).

## Discussion

Our study shows that age at definitive surgery is a significant independent prognostic factor for the persistent obstructive symptoms after pull-through, with an OR of ~ 3.4-fold for

**Table 3. Multivariate analysis of prognostic factors for persistent obstructive symptoms.**

| Factor Prognostics | *p*-value | OR (95% CI) |
|---|---|---|
| Definitive surgery (TEPT) | 0.54 | 0.75 (0.29–1.87) |
| Sex (Male) | 0.82 | 1.12 (0.41–3.09) |
| Age at surgery (<1 year) | 0.02* | 3.41 (1.18–9.91) |
| Aganglionosis type (Short) | 0.50 | 1.99 (0.26–14.96) |
| Global developmental delay | 0.60 | 1.60 (0.27–9.51) |
| Down syndrome | 0.32 | 2.63 (0.38–18.08) |

*, $p<0.05$; CI, confidence interval; OR, odds ratio; TEPT, transanal endorectal pull-through.

children who underwent pull-through at <1 year of age. Our study provides new data on prognostic factors associated with persistent obstructive symptoms following pull-through from a particular developing country. Most previous studies reported persistent obstructive symptoms after pull-through in HSCR patients from developed countries [3–5,7].

The best time to perform definitive surgery is still being debated [11–14]. Prior studies mentioned that surgery at an early age could help prevent poor bowel decompression, which happens in 25% of preoperative patients and can lead to chronic colorectal obstruction [11,12]. Contrary, surgery at a non-neonatal age provides the advantage of a more developed anal canal and sphincter complex [13]. A current meta-analysis concluded that infants under 2.5 months old show worse functional outcomes after TEPT [15]. Several possible hypotheses why younger infants have worse outcomes after pull-through have been proposed, including the pelvic floor or nerve plexus and anal sphincter prone to injury during pull-through and difficulty in identifying the normal ganglion cells in younger HSCR patients [15]. Additionally, technical approaches have been proposed as contributing factors for the outcomes [16]. Notably, the pull-through technique in our study was determined according to attending pediatric surgeons.

In this study, 21.9% of HSCR patients experienced postoperative persistent obstructive symptoms. It is similar to a previous report showing an 8–30% incidence [3]. There are three classifications of etiologies of persistent obstructive symptoms after pull-through: 1) anatomic, 2) pathological, and 3) functional (4) [3,4]. Unfortunately, we did not classify the cause of persistent obstructive symptoms in our patients. Our study focused on the prognostic factors associated with persistent obstructive symptoms following pull-through. Moreover, we did not determine the postoperative complications that might affect the persistent obstructive symptoms in our patients.

Our study also included disorders associated with HSCR, particularly global developmental delay and Down syndrome, as prognostic factors for persistent obstructive symptoms. Previous studies reported that HSCR patients with Down syndrome had worse outcomes after pull-through than those without, including persistent obstructive symptoms [17–19]. It should be noted that the number of HSCR patients with global developmental delay (6.1%) dan Down syndrome (4.4%) in our study is minimal. These facts might cause an insignificant association between those variables and persistent obstructive symptoms.

Interestingly, our HSCR patients who underwent TEPT had a ~5-fold higher possibility of having persistent obstructive symptoms than those who underwent the Duhamel procedure ($p = 0.011$). A previous study reported that among HSCR patients who suffered from persistent obstructive symptoms, 52.4%, 28.6%, and 19% underwent TEPT, transabdominal Soave, and Duhamel procedures, respectively [20]. Another report showed that 49 HSCR patients suffered from persistent obstructive symptoms, with the original pull-through of Soave (51%),

Duhamel (30.6%), and Swenson (18.4%) pull-through [21]. Persistent obstructive symptom after initial definitive surgery is usually associated with anatomic problems, with the most common etiology of anastomotic stricture [4]. Soave (endorectal) pull-through purposely leaves a muscular cuff that might develop scar tissue or roll down, causing a constricting ring and leading to mechanical obstruction [22]. These facts might be associated with the high frequency of persistent obstructive symptoms in the TEPT group in our study and other reports [20,21].

Our study noted several limitations, including 1) a retrospective design; therefore, we noted the variables, such as abdominal distension, bloating, borborygmi, vomiting, or severe constipation, relied on the medical record. Moreover, our study's retrospective nature may introduce inherent biases and limitations in data collection and analysis; 2) a single-center report, implying that more multicenter research is required to corroborate our findings. In addition, our study focuses on a single institution, which may limit the generalizability of the findings to other healthcare settings or populations; 3) our study had more patients with short segment aganglionosis and underwent TEPT; 4) the follow-up period was short, and variable among patients after the pull-through; 5) we did not use anorectal manometry as a standard tool for clinical assessment after pull-through due to tool unavailability in our hospital; and 6) we did not determine the severity of persistent obstructive symptoms experienced by our patients. Performing neonatal pull-through would be different from performing a pull-through at 3 months old (mo) *vs*. 6 mo *vs*. 9 mo of age, and simply defining age as <1 or ≥1 year old can eliminate possible differences, notably when our multivariate analysis determined that age was a significant independent factor. These aspects should be considered while interpreting our findings. In addition, we have not included enterocolitis as persistent obstructive symptoms since we have already specifically determined the prognostic factors for enterocolitis in our previous studies [23,24].

## Conclusions

Our study shows that the frequency of persistent obstructive symptoms after pull-through in our institution is considered moderate. In addition, patients who underwent pull-throughs at a younger age might have persistent obstructive symptoms following a definitive surgery. Our study provides new data on persistent obstructive symptoms after pull-through from a particular population that might be beneficial for pediatric surgeons' consideration before performing definitive surgery on patients with HSCR.

## Supporting information

**S1 Table.**
(XLSX)

## Acknowledgments

We thank those who provided excellent technical support and assistance during the study. We also thank the English editing service staff at the Faculty of Medicine, Public Health and Nursing, Universitas Gadjah Mada, for checking the manuscript's grammar. Some results for the manuscript are from Naisya Balela's thesis.

## Author Contributions

**Conceptualization:** Naisya Balela, Andi Dwihantoro,  Gunadi.

**Data curation:** Naisya Balela, Aditya Rifqi Fauzi, Ninditya Nugroho, Gunadi.

**Formal analysis:** Gunadi.

**Investigation:** Gunadi.

**Methodology:** Naisya Balela.

**Resources:** Gunadi.

**Supervision:** Andi Dwihantoro, Gunadi.

**Writing – original draft:** Aditya Rifqi Fauzi, Ninditya Nugroho, Gunadi.

**Writing – review & editing:** Aditya Rifqi Fauzi, Ninditya Nugroho, Andi Dwihantoro, Gunadi.

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
