## [Decision Letter · Decision Letter 0]

18 Jun 2023

PONE-D-23-13236Prognostic factors for persistent obstructive symptoms in patients with Hirschsprung disease following pull-throughPLOS ONE

Dear Dr. Gunadi,

Thank you for submitting your manuscript to PLOS ONE. After careful consideration, we feel that it has merit but does not fully meet PLOS ONE’s publication criteria as it currently stands. Therefore, we invite you to submit a revised version of the manuscript that addresses the points raised during the review process.

We look forward to receiving your revised manuscript.

Kind regards,

Paul Kwong-Hang Tam

Academic Editor

PLOS ONE

Reviewers' comments:

Reviewer's Responses to Questions

**Comments to the Author**

1. Is the manuscript technically sound, and do the data support the conclusions?

Reviewer #1: Yes

Reviewer #2: Yes

2. Has the statistical analysis been performed appropriately and rigorously? 

Reviewer #1: Yes

Reviewer #2: No

3. Have the authors made all data underlying the findings in their manuscript fully available?

Reviewer #1: Yes

Reviewer #2: Yes

4. Is the manuscript presented in an intelligible fashion and written in standard English?

Reviewer #1: Yes

Reviewer #2: Yes

5. Review Comments to the Author

Reviewer #1: This study examined prognostic factors for persistent obstructive symptoms in Hirschsprung disease patients following pull-through surgery in a specific developing country. The results revealed that younger age at pull-through surgery was significantly associated with a higher risk of persistent obstructive symptoms, while other factors such as operative technique and patient demographics did not show significant associations.

Strengths of the Study:

The study fills a research gap by investigating the prognostic factors for persistent obstructive symptoms in HSCR patients from a developing country, broadening our understanding of the condition in diverse populations.

The use of a cross-sectional study design and a comprehensive analysis of various factors enhance the reliability and validity of the findings.

The large sample size and inclusion of both males and females contribute to the generalizability of the results within the specific population under study.

Limitations/Issues of the Study:

The study's retrospective nature may introduce inherent biases and limitations in data collection and analysis.

The study focuses on a single institution, which may limit the generalizability of the findings to other healthcare settings or populations.

The study does not provide detailed information on the specific nature and severity of persistent obstructive symptoms experienced by the patients.

How the obstructive symptoms were handled is not demonstrated

How are the patients managed postoperatively - not demonstrated

A time to incidence of obstructive symptoms is needed

Overall, this study provides valuable insights into prognostic factors for persistent obstructive symptoms after pull-through surgery in HSCR patients from a developing country. The findings emphasize the importance of considering operative technique and age at definitive surgery when assessing the risk of persistent obstruct

Reviewer #2: Your article's goal is laudable - risk factors for Hirschsprung's associated enterocolitis and other postoperative complications after surgery are not well understood from developed countries, and data from a developing country is valuable. There are, however, a few revisions that your paper can benefit from.

1. Define "persistent obstructive symptoms" in the Introduction. This is a vague term that would benefit from clear definition as it can vary within institutions.

2. Were intraoperative frozen sections compared with permanent sections to ensure that the segment length was appropriately diagnosed? If so, how many frozen sections correlated with the permanent sections?

3. How are the postoperative persistent obstructive symptoms treated in your institution? Admission into the hospital, NPO, antibiotics, irrigations? How do these symptoms affect your patient population specifically as it may be different in developed countries?

4. While you mention that surgeon preference defined which procedures were performed, what information was used to decide when to perform a 2-stage Duhamel? Were all procedures done as 2-stage (all patients received colostomy first)?

5. What criteria was used to define short vs long segment? For total colonics, how were these reconstructed and at what age?

6. Table 1 should list how many short, long, total colonics underwent TEPT, Duhamel, or Transanal Swenson, and at what age distribution. It should also define age <1 yo better. Performing neonatal pullthrough would be different from performing a pullthrough at 3mo vs 6 mo vs. 9 mo of age, and simply defining age as less than 1 yo or greater than 1 yo can eliminate possible differences, especially when your multivariate analysis determined that age was a significant independent factor.

7. If age >1yo is associated with less obstructive symptoms, do you advocate for later pullthroughs? How are children managed prior to definitive pullthrough - colostomy vs irrigations vs other?

8. In your discussion, you mention that anorectal manometry was not used. However, were other adjuncts used to define cause of postoperative obstruction such as abdominal x-ray, contrast enema, or anorectal exam under anesthesia?

Overall, your data can be more granular to better understand potential prognostic factors, and you need to describe how persistent obstructive symptoms are treated in your institution, and what part of your practice, if any, will change based on your findings.

6. PLOS authors have the option to publish the peer review history of their article (what does this mean?). If published, this will include your full peer review and any attached files.

Reviewer #1: No

Reviewer #2: No

---

## [Author Response · Author response to Decision Letter 0]

27 Jun 2023

RESPONSES TO REVIEWER AND EDITOR:

The editor and reviewer comments are in italics in the material below, and our responses are regular.

We have ensured that our manuscript meets PLOS ONE's style requirements, including file naming.

We have uploaded our study's minimal underlying data set as Supplement File 1 (S1_File).

We have removed our ethics statement written in the section beside the Methods.

Reviewers' comments:

Reviewer 1

Reviewer #1: This study examined prognostic factors for persistent obstructive symptoms in Hirschsprung disease patients following pull-through surgery in a specific developing country. The results revealed that younger age at pull-through surgery was significantly associated with a higher risk of persistent obstructive symptoms, while other factors such as operative technique and patient demographics did not show significant associations.

Thank you very much for these encouraging comments from this reviewer.

Strengths of the Study:

The study fills a research gap by investigating the prognostic factors for persistent obstructive symptoms in HSCR patients from a developing country, broadening our understanding of the condition in diverse populations.

The use of a cross-sectional study design and a comprehensive analysis of various factors enhance the reliability and validity of the findings.

The large sample size and inclusion of both males and females contribute to the generalizability of the results within the specific population under study.

Thank you very much for these encouraging comments from this reviewer.

Limitations/Issues of the Study:

The study's retrospective nature may introduce inherent biases and limitations in data collection and analysis.

We have added the following sentences in the Discussion section: "Our study's retrospective nature may introduce inherent biases and limitations in data collection and analysis.” 

The study focuses on a single institution, which may limit the generalizability of the findings to other healthcare settings or populations.

We have added the following sentences in the Discussion section: “In addition, our study focuses on a single institution, which may limit the generalizability of the findings to other healthcare settings or populations.”

The study does not provide detailed information on the specific nature and severity of persistent obstructive symptoms experienced by the patients.

We have added the following sentences in the Discussion section: "We did not determine the severity of persistent obstructive symptoms experienced by our patients."

How the obstructive symptoms were handled is not demonstrated.

We have now added the following sentences in the Methods section: “We managed the patients with persistent obstructive symptoms as follows: first, patients were admitted into the hospital, nothing per oral, given antibiotics, and irrigations; second, defined the cause of postoperative obstructive symptoms using abdominal X-ray, contrast enema, and examination under anesthesia.3,4 Treatment of our patients depends on the cause of postoperative obstructive symptoms, whether anatomic, pathological, or functional.3,4” 

How are the patients managed postoperatively - not demonstrated.

We have now added the following section in the Methods:

Post pull-through management

The nasogastric tube was removed on postoperative day (POD) one after pull-through, while the feedings were given once the bowel sound was normal. The Foley catheter was removed on POD2 after Duhamel and TEPT, and POD4 after transanal Swenson-like pull-through. Antibiotics were given for five days after surgery. Most patients were discharged on POD5.

A time to incidence of obstructive symptoms is needed.

We have now added Supplementary File 1, including a time to incidence of obstructive symptoms. We have now also added the following sentences in the Results section: “The time to the incidence of obstructive symptoms varied among patients from 10 days to 16 months following pull-through (S1_File)”.

Overall, this study provides valuable insights into prognostic factors for persistent obstructive symptoms after pull-through surgery in HSCR patients from a developing country. The findings emphasize the importance of considering operative technique and age at definitive surgery when assessing the risk of persistent obstructive symptoms.

Thank you very much for these encouraging comments from this reviewer.

Reviewer #2: Your article's goal is laudable - risk factors for Hirschsprung's associated enterocolitis and other postoperative complications after surgery are not well understood from developed countries, and data from a developing country is valuable. 

Thank you very much for these encouraging comments from this reviewer.

There are, however, a few revisions that your paper can benefit from.

1. Define "persistent obstructive symptoms" in the Introduction. This is a vague term that would benefit from clear definition as it can vary within institutions.

In the Introduction section, we have defined persistent obstructive symptoms: "Postoperative persistent obstructive symptoms were defined as abdominal distension, bloating, borborygmi, vomiting, or severe constipation following pull-through.3”

2. Were intraoperative frozen sections compared with permanent sections to ensure that the segment length was appropriately diagnosed? If so, how many frozen sections correlated with the permanent sections?

We added the following sentences in the Methods section: “Intraoperative frozen sections were compared with permanent sections to ensure that the segment length was appropriately diagnosed. All frozen sections samples, except two cases, correlated with the permanent sections.”

3. How are the postoperative persistent obstructive symptoms treated in your institution? Admission into the hospital, NPO, antibiotics, irrigations? How do these symptoms affect your patient population specifically as it may be different in developed countries?

We have now added the following sentences in the Methods section: “We managed the patients with persistent obstructive symptoms as follows: first, patients were admitted into the hospital, nothing per oral, given antibiotics, and irrigations; second, defined the cause of postoperative obstructive symptoms using abdominal X-ray, contrast enema, and examination under anesthesia.3,4 Treatment of our patients depends on the cause of postoperative obstructive symptoms, whether anatomic, pathological, or functional.3,4” 

4. While you mention that surgeon preference defined which procedures were performed, what information was used to decide when to perform a 2-stage Duhamel? Were all procedures done as 2-stage (all patients received colostomy first)?

We have added the following sentences in the Duhamel pull-through section: "We performed a two-staged Duhamel pull-through for our patients in our hospital, i.e., all patients received stoma first.”

5. What criteria was used to define short vs long segment? For total colonics, how were these reconstructed and at what age?

We have now added the following sentences in the Methods section: “HSCR patients were classified as short-segment (aganglionosis affects the rectosigmoid colon), long-segment (aganglionosis extending proximal to the splenic flexure), and total colon aganglionosis (TCA, aganglionosis of the entire colon).”

“All TCA patients have been managed with a diverting ileostomy first. They underwent definitive surgery using the Duhamel approach at the age of at least one-year-old (Table 1).”

6. Table 1 should list how many short, long, total colonics underwent TEPT, Duhamel, or transanal Swenson, and at what age distribution. It should also define age <1 yo better. Performing neonatal pullthrough would be different from performing a pullthrough at 3mo vs 6 mo vs. 9 mo of age, and simply defining age as less than 1 yo or greater than 1 yo can eliminate possible differences, especially when your multivariate analysis determined that age was a significant independent factor.

We have now revised Table 1 as suggested by this reviewer. Moreover, we have now added the following sentences in the Discussion section: "Performing neonatal pull-through would be different from performing a pull-through at 3 months old (mo) vs. 6 mo vs. 9 mo of age, and simply defining age as <1 or �1 year old can eliminate possible differences, particularly when our multivariate analysis determined that age was a significant independent factor.”

7. If age >1yo is associated with less obstructive symptoms, do you advocate for later pull-throughs? How are children managed prior to definitive pull-through - colostomy vs irrigations vs other?

We have now added the following sentences in the Discussion section: “Performing neonatal pull-through would be different from performing a pull-through at 3 months old (mo) vs. 6 mo vs. 9 mo of age, and simply defining age as <1 or �1 year old can eliminate possible differences, particularly when our multivariate analysis determined that age was a significant independent factor. These aspects should be considered while interpreting our findings.”

We have added the following sentences in the Methods section: "Prior to definitive surgery, the children were managed by rectal irrigation. If the rectal irrigation was ineffective in keeping the colon decompressed, the children underwent colostomy.”

8. In your discussion, you mention that anorectal manometry was not used. However, were other adjuncts used to define cause of postoperative obstruction such as abdominal x-ray, contrast enema, or anorectal exam under anesthesia?

We have now added the following sentences in the Methods section: “We managed the patients with persistent obstructive symptoms as follows: first, patients were admitted into the hospital, nothing per oral, given antibiotics, and irrigations; second, defined the cause of postoperative obstructive symptoms using abdominal X-ray, contrast enema, and examination under anesthesia.3,4 Treatment of our patients depends on the cause of postoperative obstructive symptoms, whether anatomic, pathological, or functional.3,4”

Overall, your data can be more granular to better understand potential prognostic factors, and you need to describe how persistent obstructive symptoms are treated in your institution, and what part of your practice, if any, will change based on your findings.

We have made our data more granular to understand potential prognostic factors better. We have also described how persistent obstructive symptoms are treated in our institution.

---

## [Decision Letter · Decision Letter 1]

17 Jul 2023

PONE-D-23-13236R1Prognostic factors for persistent obstructive symptoms in patients with Hirschsprung disease following pull-throughPLOS ONE

Dear Dr. Gunadi,

Thank you for submitting your manuscript to PLOS ONE. After careful consideration, we feel that it has merit but does not fully meet PLOS ONE’s publication criteria as it currently stands. Therefore, we invite you to submit a revised version of the manuscript that addresses the points raised during the review process.

We look forward to receiving your revised manuscript.

Kind regards,

Paul Kwong-Hang Tam

Academic Editor

PLOS ONE

Journal Requirements:

Reviewers' comments:

Reviewer's Responses to Questions

**Comments to the Author**

1. If the authors have adequately addressed your comments raised in a previous round of review and you feel that this manuscript is now acceptable for publication, you may indicate that here to bypass the “Comments to the Author” section, enter your conflict of interest statement in the “Confidential to Editor” section, and submit your "Accept" recommendation.

Reviewer #1: All comments have been addressed

Reviewer #2: All comments have been addressed

2. Is the manuscript technically sound, and do the data support the conclusions?

Reviewer #1: Yes

Reviewer #2: Yes

3. Has the statistical analysis been performed appropriately and rigorously? 

Reviewer #1: Yes

Reviewer #2: Yes

4. Have the authors made all data underlying the findings in their manuscript fully available?

Reviewer #1: Yes

Reviewer #2: Yes

5. Is the manuscript presented in an intelligible fashion and written in standard English?

Reviewer #1: Yes

Reviewer #2: Yes

6. Review Comments to the Author

Reviewer #1: The authors present their single institution experience between 2017-2022 of patients post pull through dealing with obstructive symptoms in a developing country. They evaluated 25 patients (or 22% of the 114) to determine factors for obstructive symptoms.

The authors need to rewrite the number of patients so that there is consistent way of # of patients and percentage in all parts of the manuscript.

I still do not understand how obstructive symptoms were evaluated or why an NGT tube is required postoperatively.

They have overall provided answers to the reviewers criticisms.

Reviewer #2: (No Response)

7. PLOS authors have the option to publish the peer review history of their article (what does this mean?). If published, this will include your full peer review and any attached files.

Reviewer #1: No

Reviewer #2: No

---

## [Author Response · Author response to Decision Letter 1]

18 Jul 2023

RESPONSES TO REVIEWER AND EDITOR:

The editor and reviewer comments are in italics in the material below, and our responses are regular.

Reviewers' comments:

Reviewer's Responses to Questions

Comments to the Author

1. If the authors have adequately addressed your comments raised in a previous round of review and you feel that this manuscript is now acceptable for publication, you may indicate that here to bypass the “Comments to the Author” section, enter your conflict of interest statement in the “Confidential to Editor” section, and submit your "Accept" recommendation.

Reviewer #1: All comments have been addressed

Reviewer #2: All comments have been addressed

Thank you very much for these encouraging comments from these reviewers.

2. Is the manuscript technically sound, and do the data support the conclusions?

Reviewer #1: Yes

Reviewer #2: Yes

Thank you very much for these encouraging comments from these reviewers.

3. Has the statistical analysis been performed appropriately and rigorously?

Reviewer #1: Yes

Reviewer #2: Yes

Thank you very much for these encouraging comments from these reviewers.

4. Have the authors made all data underlying the findings in their manuscript fully available?

Reviewer #1: Yes

Reviewer #2: Yes

Thank you very much for these encouraging comments from these reviewers.

5. Is the manuscript presented in an intelligible fashion and written in standard English?

Reviewer #1: Yes

Reviewer #2: Yes

Thank you very much for these encouraging comments from these reviewers.

6. Review Comments to the Author

Reviewer #1: The authors present their single institution experience between 2017-2022 of patients post pull through dealing with obstructive symptoms in a developing country. They evaluated 25 patients (or 22% of the 114) to determine factors for obstructive symptoms.

The authors need to rewrite the number of patients so that there is consistent way of # of patients and percentage in all parts of the manuscript.

We have now rewritten the number of patients so that there is a consistent way of # of patients and percentage in all parts of the manuscript.

I still do not understand how obstructive symptoms were evaluated or why an NGT tube is required postoperatively.

We have now revised the Methods section: “After admission, the patients with persistent obstructive symptoms were managed with antibiotics, irrigations, and nothing per oral. They were evaluated using the following diagnostic approach: history, abdominal X-ray, contrast enema, and examination under anesthesia.3,4 Treatment of patients was determined according to the etiology of the obstructive symptoms, anatomic, pathological, or functional.3,4” 

The placement of NGT was our protocol from the anesthesiologist during the surgery. The NGT was removed soon after the surgery.

We have now revised the Methods section:

“Post pull-through management

The nasogastric tube was removed soon after surgery finished.”

They have overall provided answers to the reviewers criticisms.

Thank you very much for these encouraging comments from this reviewer.

Reviewer #2: (No Response)

7. PLOS authors have the option to publish the peer review history of their article (what does this mean?). If published, this will include your full peer review and any attached files.

Do you want your identity to be public for this peer review? For information about this choice, including consent withdrawal, please see our Privacy Policy.

Reviewer #1: No

Reviewer #2: No

---

## [Editor Report · Decision Letter 2]

9 Aug 2023

Prognostic factors for persistent obstructive symptoms in patients with Hirschsprung disease following pull-through

PONE-D-23-13236R2

Dear Dr. Gunadi

We’re pleased to inform you that your manuscript has been judged scientifically suitable for publication and will be formally accepted for publication once it meets all outstanding technical requirements.

Kind regards,

Paul Kwong-Hang Tam

Academic Editor

PLOS ONE

Additional Editor Comments (optional):

Reviewers' comments:

Revision is satisfactory

---

## [Editor Report · Acceptance letter]

30 Aug 2023

PONE-D-23-13236R2 

Prognostic factors for persistent obstructive symptoms in patients with Hirschsprung disease following pull-through 

Dear Dr. Gunadi:

I'm pleased to inform you that your manuscript has been deemed suitable for publication in PLOS ONE. Congratulations! Your manuscript is now with our production department. 

Kind regards, 

on behalf of

Dr. Paul Kwong-Hang Tam 

Academic Editor

PLOS ONE